# Fingerprinting the Shadows: Unmasking Malicious Servers with Machine Learning-Powered TLS Analysis

## ABSTRACT

Over the last few years, the adoption of encryption in network traffic has been constantly increasing. The percentage of encrypted communications worldwide is estimated to exceed 90%. Although network encryption protocols mainly aim to secure and protect users' online activities and communications, they have been exploited by malicious entities that hide their presence in the network. It was estimated that in 2022, more than 85% of the malware used encrypted communication channels.

In this work, we examine state-of-the-art fingerprinting techniques and extend a machine learning pipeline for effective and practical server classification. Specifically, we actively contact servers to initiate communication over the TLS protocol and through exhaustive requests, we extract communication metadata. We investigate which features favor an effective classification, while we utilize and evaluate state-of-the-art approaches. Our extended pipeline can indicate whether a server is malicious or not with 91% precision and 95% recall, while it can specify the botnet family with 99% precision and 99% recall.

## KEYWORDS

TLS, TLS Fingerprinting, Active Probing, Botnet, Command and Control, Server Characterization, Machine Learning

**ACM Reference Format:**

Anonymous Author(s). 2023. Fingerprinting the Shadows: Unmasking Malicious Servers with Machine Learning-Powered TLS Analysis. In *Proceedings of Make sure to enter the correct conference title from your rights confirmation emai (1).* ACM, New York, NY, USA, 14 pages. https://doi.org/10.1145/nnnnnnn.nnnnnnn

## 1 INTRODUCTION

As of October 2020, more than 90% of Internet traffic is communicated over TLS [4]. The adoption rate of TLS 1.3 surpassed the adoption rate of previous versions of TLS with remarkable speed. Based on the TLS Telemetry Report in 2021 [9], TLS 1.3 has become the preferred protocol for 63% of the top 1 million web servers on the Internet[3]. Yet, the increasing popularity and simplified usage of TLS led malware to exploit encryption to hide its presence and communications [13]. WatchGuard observed in 2021 that 91.5% of malware is delivered through encrypted channels [12]. Furthermore, the TLS Telemetry Report revealed that the proportion of phishing sites using HTTPS and valid certificates had risen to 83% in the same year [9].

To establish a secure channel using the TLS protocol, a crucial step involves the exchange of metadata between the client and the server. This metadata is shared through unencrypted *Client Hello* and *Server Hello* messages. Previous work [2, 8, 28, 39] has acknowledged the significance of metadata exchanged during the TLS handshake and attempted to leverage this information to enhance Internet security [45]. These works focus on the passive analysis of the messages exchanged during the TLS handshake.

By employing active approaches, like active TLS fingerprinting, researchers and organizations [16, 22] can collect relevant TLS parameters that when properly processed, they enable the extraction of valuable information for the network status and the participating devices and applications [7, 44].

In this work, we leverage a range of active TLS fingerprinting techniques to examine server behavior. First, we initiate our interaction with servers by sending the first request from a set of predefined requests, and then, we start re-sending this request by progressively forcing the server to select an alternative cipher suite. These TLS parameters then are utilized from the enhanced version of a publicly available semi-automatic machine learning pipeline [21]. To ensure a robust evaluation, we employ three distinct classification models. Building upon this methodology, we compile a database with traffic from benign and malicious entities. We create a robust binary classification system capable of accurately labeling servers as either benign or malicious. Additionally, we implement a multi-class classification model to identify specific botnet families. Finally, we implement a TLS fingerprinting technique and evaluate the performance of 4 different categories. Through this comparison, we explore the strengths and limitations of each method. The contributions of this work follow:

- We examine state-of-the-art approaches for server classification through active probing and integrate them into a publicly available machine learning pipeline.
- We present the most extensive dataset known to date for actively categorizing servers, containing information from all current approaches. Upon acceptance of the paper we will make the dataset publicly available[1].
- We present an analysis of these approaches using a comparative evaluation between (i) 4 different features categories based on fingerprinting and (ii) 3 machine learning models for Binary and Multi-Class classifications.

## 2 BACKGROUND

To establish a secure channel via TLS, a secure TLS parameters exchange is essential immediately following the TCP handshake, allowing both sides to share their capabilities and preferences.

The TLS handshake begins with the client sending a *Client Hello* message to the server. This message contains crucial information, such as the supported TLS versions, cipher suites, and extensions. It serves as a way for the client to communicate its capabilities and preferences to the server. Upon receiving the *Client Hello* message, the server responds with a *Server Hello* message. This message includes details about the selected TLS version, cipher suite, and

---

[1]The dataset includes data from the top 10K Tranco domains [15], 1 labeled [10] and 4 unlabeled blocklists [11, 14, 18, 19] over a 5-month period of daily measurements (01/2023 – 05/2023).

other parameters chosen by the server. It allows the server to inform the client of its preferences and capabilities, enabling both parties to negotiate and agree upon the most secure and compatible settings for their communication.

A common technique is to leverage these parameters during the initial stages of establishing the secure channel for different purposes (e.g., OS identification [44], server-side libraries [43], client identification [37, 50], or in censorship circumvention tools [36]), since the messages exchange happens in plaintext. TLS incorporates various parameters and configurations that contribute to its security and compatibility. The cipher suite, for example, determines the encryption algorithms and key exchange methods used for secure communication.

TLS fingerprinting and Machine Learning utilize the parameters extracted from the *Server Hello* messages during the TLS handshake to perform server identification and characterization. By analyzing the unique combination of these parameters, ideally, a distinguishing identifier is generated for each different configuration, enabling the classification and differentiation of servers.

## 3 RELATED WORK

Although we follow an active probing approach, in this paragraph, we briefly discuss related works that focus on passive analysis. In 2009, Ristic's work [52] made TLS fingerprinting gain popularity [27], leading to more research [1, 2, 28, 32, 39, 53]. Over time, TLS fingerprinting was applied to diverse applications. It has been employed for longitudinal studies [43], instrumental in distinguishing android apps [50], and even used in discerning IoT ecosystem components[49].

In the domain of active probing-based fingerprinting techniques, there are two popular methodologies, ATSF [58] and JARM [7], which send a fixed number of *Client Hello* messages to a server. Both techniques generate fingerprints according to the *Server Hello* responses they receive. Papadogiannaki et al. [48] use JARM to generate fingerprints for servers that is known to participate in botnets and they examine their evolution over time. They also show that the percentage of fingerprint overlapping with benign servers progressively rises. In ATSF, authors propose a list of alternative *Client Hello* messages, which seem to result to more expressive and optimized fingerprints when compared to JARM. Nonetheless, a notable aspect of these approaches is the initial generation of the first 10 *Client Hello* messages. This initial step, could benefit from heightened explainability, thus facilitating potential optimization strategies and refining the generation process. DissecTLS [57] introduces an enhanced functionality of a recursive process of systematically excluding selected server parameters or preferences from the interaction sequence. By iteratively eliminating these values and subsequently re-sending the requests, the DissecTLS methodology strives to generate more expressive fingerprints. Notably, DissecTLS concentrates on the comprehensive extraction of server configurations for server classification. This emphasis on configuration extraction sets DissecTLS apart, offering insights into server behavior outperforming similar tools [24, 25].

To the best of our knowledge, there is no related work that uses machine learning to classify server activity with information collected after active probing. Although several studies have explored the utilization of machine learning techniques in conjunction with TLS parameters and fingerprinting [31, 38, 40, 42, 44, 47, 51, 55, 60], they focus on different use cases (e.g., OS identification, website fingerprinting, detection of privacy leaks). To accurately compare works that do not share analogous techniques is challenging, since the combination of active probing and machine learning adds an additional layer of complexity. Kim et al. [41] remarkably focus on analyzing TLS traffic based on enhanced neural networks combined with TLS fingerprinting methods. However, its reliance on passive techniques contrasts with our emphasis on active ones.

In this work, we aim to explore every facet of existing approaches, striving to identify the optimal outcome in terms of classification. That's why we employ active techniques to gather a broad range of features and subsequently feed them into machine learning models, which are then utilized in our fingerprinting methods. In the upcoming sections, we'll explore the benefits of fingerprinting methods used in active probing and combine the different techniques used in ATSF, JARM and DissecTLS. Finally, we use a modified version of a publicly available machine learning pipeline tailored for our experiments. By testing different fingerprinting techniques and adapting the machine learning models, we aim to examine how they perform in active scenarios.

## 4 METHODOLOGY

In this section, we present an overview of the methodology used in this study. Our approach centers on optimizing and comparing the current active approaches of collecting unique messages from servers during TLS parameter negotiation.

An approach to attain this is by employing a predefined number of particular requests to a server and receiving the corresponding responses in return. For this purpose, instead of generating new initial *Client Hello* message, we leverage existing ones from previous works. We select the JARM tool, renowned for its scalability and efficiency [6], widely adopted by major Internet scanners, such as *Censys* [17] and *Shodan* [23]. JARM sends 10 customized TLS *Client Hello* messages to a target TLS server, enabling us to identify a distinct set of responses.

Each one of these *Client Hello* messages triggers a response from the server in the form of a *Server Hello* message, agreeing on the selected parameters. During this process, it is possible for a server to abort the handshake or attempt to renegotiate. DissecTLS [57] exploits this opportunity by conducting intensive scans on servers. This process continues until the complete TLS configuration can be successfully reconstructed. Based on this approach, we implement an extension of JARM tool which integrates this additional functionality. Our tool executes each of its initial requests by progressively removing previous preferences. Our choice is to concentrate on a specific parameter for reduced complexity. The key is to iteratively remove previous cipher suites selected by servers. This process continues until an error occurs or a timeout elapses. This proves to be the best case due to its wide range of options and the presence of a preference order. More details can be found in §4.1. Finally, we utilize and extend an existing pipeline and proceed to enhance and fine-tune it for the purpose of server classification through active probing.

## 4.1 Cipher Suite Selection

To enhance the method of active probing and achieve a more streamlined process, we focus on a single parameter during TLS parameter negotiation – the cipher suite. The cipher suite plays a crucial role in establishing a secure channel between the client and server. Cipher suites encompass a wide variety of cryptographic algorithms and key exchange methods, and their order of preference can significantly impact the server's behavior.

We perform an iterative process, where we interact with the server by initiating TLS connections and start removing the selected cipher suites one by one. We continue this process until an error occurs or a timeout is reached. The primary objective is to identify the server's preferred cipher suite by observing its behavior when specific cipher suites are eliminated. By concentrating on a single parameter, we aim to extract all possible preferences of the server related to the cipher suite, leading to more unique responses. This approach not only increases the precision of our model, but also reduces the extra overhead associated with probing multiple parameters simultaneously, while also delaying the occurrence of errors. We monitor the server's responses and gather data on each interaction. This data serves the crucial purpose of identifying the server's most preferred cipher suite and comprehending how its behavior evolves when different cipher suites are removed.

## 4.2 Data Collection

Our data collection methodology is carefully designed to encompass a wide spectrum of servers and network configurations. The process involves two main sources: the "Top 10K domains" from the Tranco list and various "Blocklists" containing potentially malicious IP addresses.

*4.2.1 Top 10K Domains.* To initiate our data collection process, we select the "Top 10K domains" from the Tranco list, which is a reliable source of Internet-wide domain information [15]. Considering the time and resources required to scan the entire Tranco list on a daily basis, we scan a representative sample; the top 10K domains. To maintain the relevance and timeliness of our data, we schedule nightly downloads. This regular update process allows us to capture evolving trends within the TLS ecosystem.

*4.2.2 Blocklists.* In addition to the top 10K domains, we strategically incorporate the Feodo Blocklist into our data collection [10]. This blocklist offers a unique opportunity to examine IP addresses associated with five different botnet families. To expand our dataset, we also include unlabeled blocklists that contain potential malicious IP addresses (i.e., Blocklists.de [14], Ci-Badguys [18], SSLBL [11], and Darklist.de [19]). *Blocklists* term in the rest of the paper refers to these unlabeled lists. The daily number of unique IP addresses contained in these blocklists ranges from 30 to 20K. This integration enables us to gain insights into existing TLS behaviors across a broader range of servers, encompassing both benign and potentially malicious activities. To handle the large size of blocklists, we employ weekly Censys's active TLS scans [33], which filter the lists and provide us with an up-to-date and manageable set of active IP addresses along with a list of open ports.

*4.2.3 Database.* After this preparation phase, we employ our tool, starting to send its 10 consecutive *Client Hello* messages. For each one of them, we proceed with an iterative process, where we remove the cipher suite selected by the server. Initially, we send the first *Client Hello* message to the server and wait for its response. Upon receiving the server's selection of a cipher suite, we proceed with removing this specific option from the list of supported cipher suites for the subsequent interaction. We then resend the same *Client Hello* message to the server, this time without the eliminated cipher suite. This iterative approach continues until the server either refuses to respond or stops acknowledging our queries. Subsequently, we repeat this entire procedure for each of the remaining 9 *Client Hello* messages in a methodical manner.

Through this process, we acquire valuable insights into TLS parameter negotiation, allowing us to assess how servers dynamically adapt their responses to different cipher suite options. Throughout this interaction, we monitor and collect the network packets exchanged using the `tcpdump` tool, saving the data into packet capture files. In contrast to real-time fingerprint generation approaches, we store the collected traffic for subsequent analysis. This decision allows us to focus on the efficiency of feature selection when creating fingerprints. The stored traffic forms a rich dataset that enables deeper investigations into TLS parameter negotiation and server behavior.

Over a five-month period (01/2023 – 05/2023), we accumulate approximately 1.8M samples, resulting in a database totaling to 278GB in size. This extensive repository empowers us to observe evolutionary trends within the TLS ecosystem, monitor how malicious botnets adapt their activities on benign servers, and evaluate the effectiveness of various state-of-the-art approaches. Furthermore, this dataset provides fertile ground for exploring novel capabilities and potential in TLS parameter negotiation and secure channel establishment research.

## 4.3 Filtering

To ensure the integrity and reliability of our dataset, we implement a filtering process to retain only the successful samples while reducing potential anomalies. First, we remove servers that did not respond to any of the initial 10 *Client Hello* messages, resulting in empty pcap files.

Next, we identify scenarios where servers initially respond successfully to our requests but subsequently stop acknowledging our messages during the TLS renegotiation, leading to a timeout. Such occurrences are flagged as "Incomplete", as they do not provide valuable insights into server behavior in cases that communication is unexpectedly disrupted. Furthermore, we encounter instances that servers respond with no SSL/TLS packets or repeatedly provide the same *Server Hello* for each request. These anomalies are marked as "Disrupted", and we discard them from processing alongside with "Incomplete" samples.

To minimize these anomalies to the maximum extent possible, as a final step, we perform a flow checksum on the collected traffic. We carefully check and confirm that the way each communication happens follows the usual patterns and rules expected. For example, we make sure that the TLS information is contained within the TCP packet. We also look at cases where packets arrive too late (after the timeout lapsed) or when ACK numbers do not match the SEQ ones. If things don't match up or follow the rules, we remove that

**Table 1: Overview of the final number of data samples processed.**

| Source | Filtered Samples |
|---|---|
| Tranco | 763,443 |
| Blocklists | 84,354 |
| QakBot (Feodo) | 3,890 |
| Dridex (Feodo) | 1,369 |
| BumbleBee (Feodo) | 931 |
| Emotet (Feodo) | 863 |
| BazarLoader (Feodo) | 75 |
| Total | 854,925 |

data from our collection. Table 1 provides an overview of the final data samples resulting from our filtering process.

## 4.4 Data Transformation and Parameters Selection

After filtering our dataset and retaining only the *Completed* files, the next crucial step is to address the challenge posed by the size of packet capture files, since processing and feeding them directly into a classification model for training proved impractical. Therefore, we have created a lighter format that would still retain the essential features needed for our analysis and evaluation. To achieve this, a subset of TLS parameters, extensions, and certificates is carefully selected[2]. These parameters include TLS versions, cipher suites, ALPNs, Elliptic Curves, certificates (x509) and other relevant information. During the data transformation phase, we aim for a balance between information richness and efficiency. Thus, we create a list large enough to enable the utilization of different parameters subsets and perform comparisons effectively. Finally, we unify the original data samples into a single CSV (Comma-Separated Values) file, containing all the extracted fields and reaching the size of 52 GB. Its simplicity and ease of use make it an ideal choice for representing the curated list of parameters and TLS negotiation data.

To ensure consistent data presentation, each packet capture file in our database corresponds to a single row into a CSV file. This row contains all the parameters extracted from each server's response, providing a total view of the TLS negotiations that occur. Each column holds a particular parameter's value that the server chose from the corresponding request at that specific time. Since we perform exhaustive requests to servers, the exact number of their total responses for each *Client Hello* varies, resulting in a dataset that includes rows with features ranging from 200 to 20K. The calculation yielding 20K features is determined by the following factors: we extract 46 distinct parameters from each *Server Hello* message, the maximum observed number of iterations is 45, and there are 10 initial handshake procedures. Consequently, if a server responds 45 times recursively for each of the 10 initial handshakes, the resultant dataset comprises 20,000 features (46 parameters x 45 iterations x 10 handshakes), in addition to columns indicating the corresponding date of the sample, category, botnet family (if applicable), IP/domain, port, and 10 columns indicating the total

---

[2]We present these parameters in Table 8, included in the Appendix

number of responses for each of the 10 initial handshakes until the server encounters an error or fails to respond.

Finally, in the last step, we transform string-type values into an arithmetic-like format representation. This conversion is necessary for the selected classification machine learning models, as they can only process numeric features. We implement a technique to represent strings as integers while ensuring consistency for the same input. By leveraging the MD5 hash function, we effectively convert these string-based parameters into a standardized format that is suitable. Consequently, we successfully transform the string-based parameters into fixed-size representations and then convert the hexadecimal output into integers. In cases where the input is empty, for example, 'None', our implementation returned '-1' to maintain data integrity and signify the absence of applicable data.

## 5 ANALYSIS

Before proceeding to the design of our models, we conduct a preliminary analysis on the final transformed and extracted data to gain insights and understand the variation in server responses when performing exhaustive requests.

In the analysis phase, we calculate the average number of responses for each data source to examine the servers behavior during TLS parameter negotiation and cipher suite selection. Additionally, we apply the Borda Count rule [34, 35, 54] to identify the top cipher suites among the servers in our dataset, based on their min-max normalized scores [26].

To gain a deeper understanding of the data, we visualize the distribution of server responses. Following figures represent the average responses of the sources and the distribution of them through box plots, provide an overview of the data central tendency, dispersion, and outliers. By analyzing these plots, we examine the variation in server responses and determine if exhaustive techniques can lead to better results. The insights gained from this preliminary analysis help us identify patterns, trends, and potential variations in server behavior during TLS parameter negotiation. Understanding these aspects is critical to move on with the evaluation phase and compare of our models. The box plots, in particular, allow us to visually compare the distribution of server responses across different sources, highlighting any variations and enable us to draw important conclusions about the impact of uniqueness in server classification.

## 5.1 Number of Responses

Figure 1 presents the mean number of responses for each source (i.e., Tranco, Blocklists, Feodo) during the 10 exhaustively iterative handshakes until the server either responds with an error message or ignores the request. We observe that at the last handshakes the majority of sources exhibit lower number of responses compared to earlier ones. This could open up intriguing possibilities for future investigations into the specific *Client Hello* parameters responsible for this reduction and why they influence server behavior. Dridex and Emotet botnets exhibit the highest average number of responses among all sources. Notably, the Dridex botnet demonstrates a high response rate across all handshakes, indicating its reluctance to refuse a TLS connection regardless the utilization of any outdated parameters. On the other hand, the Emotet botnet seems to respond


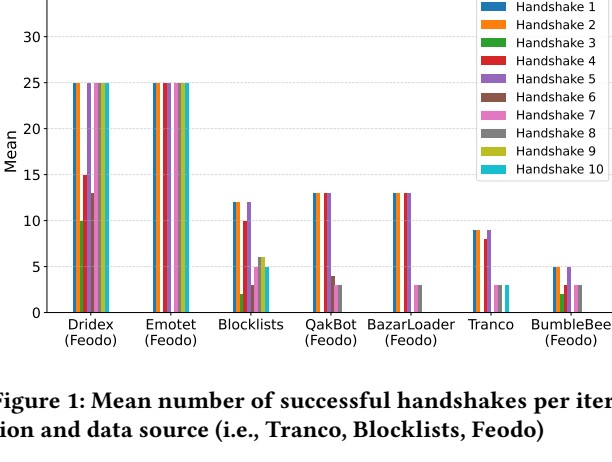

**Figure 1: Mean number of successful handshakes per iteration and data source (i.e., Tranco, Blocklists, Feodo)**

in a standardized manner, consistently producing specific patterns in its responses following a specific configuration.

Figure 2 illustrates the distribution and variability of the total response counts across the different sources. The extent of spread within the boxes and the length of the whiskers directly indicate the degree of variation. *Tranco*, which represents benign server activity, exhibits a high number of outliers. This could be because of certain servers that might not show up frequently in the Tranco list, yet they have configurations leading to high response rates. Another reason could be the presence of limited servers with responses significantly higher from the majority. This outcome is in line with expectations considering the extensive diversity among domains and their unique configurations. In contrast, the *Feodo* source has a reduced number of outliers (resulting from BumbleBee, Emotet and QakBot botnet families). It consists traffic from 5 different botnet families, each consistently following its distinct behavior. For this source, an outlier might indicate a misclassification. Lastly, interpreting the list namely *Blocklists* is challenging, since the family of the IP addresses contained is unknown. Unlike the *Feodo* list, the list *Blocklists* contains raw IP addresses with no other information.

## 5.2 Variances in Cipher Suite Selection

In the next phase of our analysis, we evaluate the cipher suite preferences exhibited by various sources within the dataset. We extract the cipher suite selections for each *Client Hello* message and aggregate them based on the specific handshakes employed. To benchmark the preferences, we harness the Borda Count rating technique [35]. This method assigns points to candidates according to their rankings on each ballot. Lower-ranked candidates are allocated fewer points, while higher-ranked ones garner more. By summing up the scores granted to each cipher suite (divided per source list), we derive server preferences. To ensure uniformity, we apply min-max normalization. This allows us to identify the cipher suite selections per source list, based on the initial handshakes and to observe variations across sources. Coupled with the analysis of response counts, these findings could potentially reveal botnet attempts to imitate benign TLS configurations. Table 2 presents the top 10 Tranco cipher suites selection. For each cipher suite, we

**Table 2: Top 10 SSL/TLS cipher suites of Tranco Ranked by Borda count (presented in descending order) and the corresponding values of the C2 lists**

| Cipher | Score | | |
|---|---|---|---|
| | Tranco | Blocklists | Feodo |
| TLS_ECDHE_RSA_WITH_AES_128_GCM_SHA256 | 1.000 | 0.998 | 0.975 |
| TLS_ECDHE_RSA_WITH_AES_256_GCM_SHA384 | 0.965 | 1.000 | 0.964 |
| TLS_ECDHE_RSA_WITH_AES_128_CBC_SHA256 | 0.742 | 0.730 | 0.856 |
| TLS_ECDHE_RSA_WITH_AES_128_CBC_SHA | 0.730 | 0.834 | 1.000 |
| TLS_ECDHE_RSA_WITH_AES_256_CBC_SHA384 | 0.711 | 0.756 | 0.843 |
| TLS_ECDHE_RSA_WITH_AES_256_CBC_SHA | 0.703 | 0.846 | 0.979 |
| TLS_RSA_WITH_AES_128_GCM_SHA256 | 0.586 | 0.650 | 0.751 |
| TLS_RSA_WITH_AES_128_CBC_SHA | 0.563 | 0.940 | 0.873 |
| TLS_RSA_WITH_AES_256_CBC_SHA | 0.531 | 0.937 | 0.852 |
| TLS_RSA_WITH_AES_256_GCM_SHA384 | 0.497 | 0.658 | 0.737 |
| Unique cipher suites | 69 | 68 | 49 |

display its corresponding score as occurred during the examination of the Blocklists and Feodo source lists.

## 6 MODEL DESIGN

In this section, we present the machine learning classification approach and an implementation of a fingerprinting method with 4 different feature categories, each one derived from a different strategy. Our aim is to explore the potential of these methods for server classification. Notably, there is a significant gap in the existing literature, as no prior work has delved into the use of machine learning models for server classification with active probing. Our research seeks to address this limitation by investigating the effectiveness of machine learning models in this unique context, combining the benefits of active probing.

## 6.1 Machine Learning Model

In this study, one of our main objectives is to investigate the efficacy and accuracy of machine learning classification approaches. We examine the characterization of servers into benign and malicious (binary classification), as well as the categorization into different types of malicious sources (multi-class classification). Machine learning classification tasks entail several key steps, including data pre-processing, feature selection, defining multiple models along with their configurations, fine-tuning during cross-validation, and ultimately selecting the best model. It is crucial to execute each step properly to avoid common pitfalls such as information leakage between training and testing sets, class imbalance, overfitting/underfitting, and decision threshold optimization.

To ensure a robust implementation, we have chosen a publicly available semi-automatic machine learning pipeline that addresses common issues encountered in classification tasks [21, 56]. Furthermore, we have selected 3 distinct classification models to facilitate a comprehensive comparison of different classification approaches: Gaussian Naive Bayes [59], Random Forest [29], and the state-of-the-art XGBoost (eXtreme Gradient Boosting) [30]. These selected methods are well-equipped to handle class imbalance and perform effectively on imbalanced datasets, a crucial consideration for our dataset.

The chosen pipeline includes data division into 80%/20% proportions for training-validation and testing (also known as hold-out). However, this random split based solely on the sample class category (also known as a target) is not suitable for our dataset due to

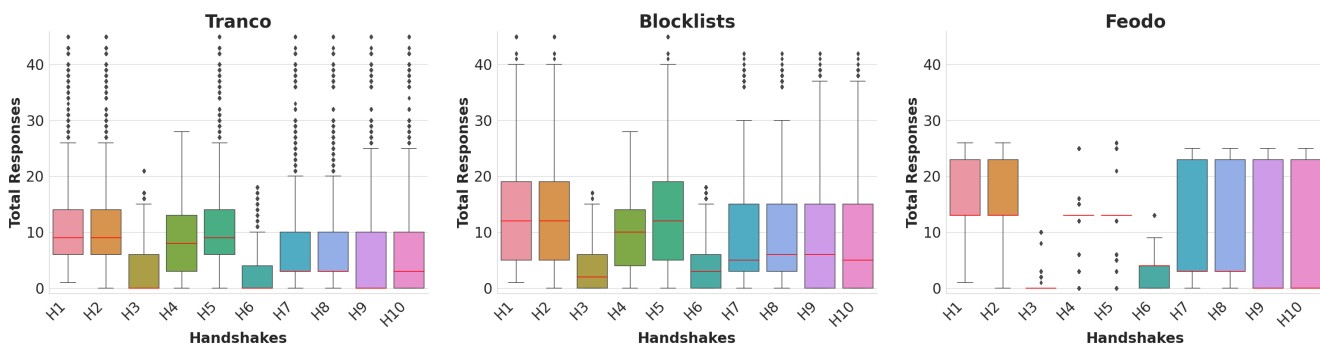

Figure 2: The distribution of total responses across the sources (i.e. Tranco, Blocklists, Feodo)

the presence of different malicious or benign machines that have been sampled multiple times (daily measurements). In some cases, a machine may or may not change its configuration during the monitoring period, and the initial splitting based solely on the target value could result in nearly identical samples being present in both the training and testing data portions. This type of data separation could lead to overly optimistic model performance.

To address this issue, we have taken measures to ensure the appropriateness of our data splitting. Specifically, we have retained only unique samples for each machine in our dataset. Furthermore, we have divided the dataset into 80% for train/validation and 20% for the testing portion. This division is based on the combination of the target (benign or malicious) and the machine's IP address, ensuring that the testing set contains 20% of the unique machines, avoiding the presence of similar samples between training and hold-out set.

## 6.2 Fingerprinting

Fingerprinting techniques consist two fundamental steps. The first step focuses on feature selection, which is the most significant aspect of this methodology. During this phase, most informative and discriminative parameters from TLS metadata are carefully curated and identified. The selected features play a pivotal role in characterizing the server behavior and for effective classification. The second step involves the fingerprint generation by concatenating these features and then by utilizing hash functions, convert them into standardized format outputs.

In a dataset with more than 20K possible features, selecting the optimal ones is challenging. Based on current approaches we create 4 distinct combinations of features. Each category offers valuable insights into the trade-off between the selected features and the model's precision. Table 3 presents the different categories. *Exhaustive* category, contains features from only 1 parameter, the chosen cipher suite extracted from each *Server Hello* during the handshakes. Although changing cipher suites could impact other parameters, we concentrate on the most relevant one during our step-by-step removal process. Conversely, in *Predefined* category, 21 out of the 46 parameters are included as features as we choose to exclude parameters relevant to certificates (x509af_version, x509af_serialnumber etc.) and instead concentrate solely on behavioral patterns. These

Table 3: The maximum number of selected features/unique parameters extracted from the corresponding successful iterations per category

| Exhaustive | Predefined | ML-Selected | All-Possible |
|---|---|---|---|
| 450 | 210 | 84 | 20710 |

features are exclusively extracted from the initial response of each handshake, ignoring the previous exhaustive approach completely.

The last 2 categories are designed with the first containing the features selected from the machine learning pipeline outlined in §6.1 and the second with all features possible, with an additional 10 features that indicate the total responses per handshake.

To generate the fingerprints, we follow a simple approach. We concatenate the selected features and pass directly through the *SHA-256* hash function, resulting to a 32-byte output. This process should provide a single fingerprint for each unique server behavior.

JARM's 62-character fingerprints are segmented, with the initial 30 characters representing the server's chosen TLS version and ciphers for each of the 10 client hello messages, and the subsequent 32 characters forming a truncated SHA256 hash of the cumulative server extensions, excluding x509 certificate data [20]. Unlike JARM, we're not emphasizing in a fingerprint generation process that relies on partial fingerprint matching for similarities. Using the methodology described above, our goal is to observe the variations in results across the different feature categories, each representing a different feature selection approach.

Finally, we use a different approach for data splitting compared to the machine learning process. Given that fingerprinting techniques depend on periodically updated databases we divide the dataset into 80%-20% based on the dates collected. We retain the traffic from the initial 80% of dates for training, generating fingerprints that are stored in our database, while the remaining 20% is allocated for testing.

## 7 EVALUATION

In this section, we outline the various steps we followed to evaluate our models. The entire process was carried out on a machine with 512GiB of system memory, driven by 2 AMD EPYC 7543 32-Core

**Table 4: Number of total and unique fingerprints per feature category (i.e., *Exhaustive, Predefined, ML-Selected, All-Possible*) across the different data sources.**

| Source | Total FPs | Unique FPs | | | |
|---|---|---|---|---|---|
| | | Ex/ve | Pred/ed | ML | All |
| Tranco | 763K | 5K | 123K | 207K | 329K |
| Blocklists | 84K | 2K | 19K | 27K | 35K |
| QaKBot (Feodo) | 3K | 4 | 109 | 354 | 1855 |
| Emotet (Feodo) | 863 | 2 | 37 | 72 | 158 |
| Dridex (Feodo) | 1369 | 3 | 42 | 62 | 330 |
| BumbleBee (Feodo) | 931 | 5 | 16 | 35 | 215 |
| BazarLoader (Feodo) | 75 | 3 | 28 | 3 | 33 |
| Total | 854K | 7K | 143K | 235K | 367K |

Processors, and equipped with an Nvidia A30 GPU. This configuration was essential for efficiently handling and processing the substantial amount of data, particularly given the extensive number of extracted features and the fine-tuning of machine learning models.

We evaluate our models based on two distinct classification techniques. First, we perform binary classification to specify whether a server is malicious (C&C) or not. Then, we perform multi-class classification, where we focus on the labeled traffic collected from *Feodo*, to identify the botnet family of each sample. To compare the performance of our models, we extract precision, recall and F1 metrics.

## 7.1 Pre-Assessment

Before training, we perform some checks for possible IP address overlaps over the whole dataset. We encounter only 3 IP addresses overlapping in different dates between the *Emotet* and *Dridex* botnets of Feodo. For realistic reasons, none of them was excluded.

Subsequently, we extract all possible fingerprints from our different feature selections. Table 4 provides an overview of the total and unique fingerprints per category. Interestingly, even though *ML-Selected* category contains the fewest features, produces more unique fingerprints than the *Exhaustive* and *Predefined*.

In our next step, we aim to determine the number of overlaps between the fingerprints. Figure 3 illustrates the unique overlaps across sources. We observe that, once again, the *ML-Selected* category, despite having fewer features, exhibits fewer overlaps compared to the *Exhaustive* and *Predefined*. Notably, several servers from the *Blocklists* list share identical fingerprints with the *Tranco*, possibly indicating an effort to imitate legitimate behavior and duplicate configurations.

## 7.2 Malicious vs benign

*7.2.1 Machine Learning.* As previously indicated, the binary classification pipeline effectively reduces the dataset's dimensionality by selecting only the top 84 most important features from the pool of maximum 20K extracted features. We identify the significant features using the Lasso regression model based on non-zero coefficients. Following the feature selection process, the pipeline proceeds to fine-tune each predefined model, retaining only the

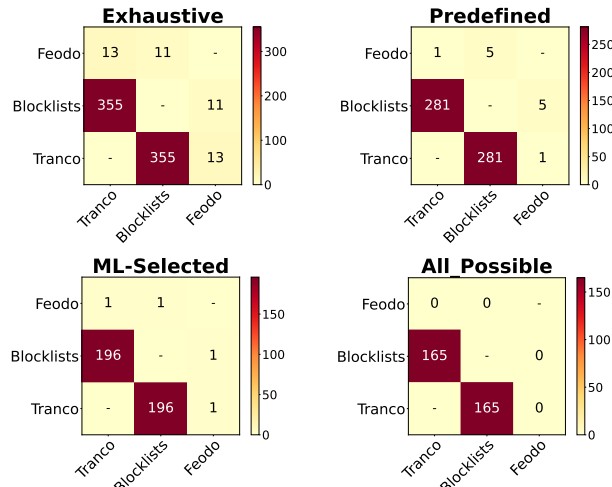

**Figure 3: Number of overlaps for each pair of sources (unique)**

best configuration for each. The average performance of these configurations during K-Fold cross-validation is summarized in Table 5. The results strongly indicate that XGBoost is the optimal classification configuration, achieving an average F1 score of 0.973 over the validation set.

To comprehensively evaluate the performance of the selected model, we utilize the remaining of the hold-out dataset, which contains 120,374 unseen data samples. The chosen configuration achieves an F1 score of 0.931, a ROC-AUC score of 0.971, a recall score of 0.95, and a precision score of 0.911. Additionally, to further evaluate and stress the selected model, we repeat the testing procedure with newly added data, originating from IP addresses never processed before[3]. Specifically, we collect different sets of unseen benign and malicious machines for 2 separate time periods (06/2023 and 07/2023). We process unseen data from 18,883 samples collected in June and 21,782 samples collected in July. We utilize these data samples in order to evaluate the model performance in a realistic case scenario. According to the results presented in Table 6, our model (XGBoost) manages to achieve more than 0.95 F1 score and more than 0.986 precision in both datasets (06/2023 and 07/2023).

**Table 5: Performance of the selected binary and multi-class classification models during the K-Fold cross validation, measured in F1 score**

| Model | Binary | | Multi-class | |
|---|---|---|---|---|
| | Training | Validation | Training | Validation |
| XGBoost | **0.986** | **0.973** | **0.997** | **0.981** |
| Random Forest | 0.948 | 0.939 | **0.997** | **0.981** |
| Gaussian NB | 0.863 | 0.863 | 0.930 | 0.925 |

In addition, we have extracted the explanation of the captured model patterns using the SHAP model explainability technique.[4]

---

[3]Same sources used: 10K Tranco domains [15], Feodo [10] and blocklists [11, 14, 18, 19]
[4]Presented in Figure 6, included in the Appendix

**Table 6: Performance of the selected binary classification model over three different datasets: (i) the hold-out dataset (120,374 samples), (ii) the dataset collected in June 2023 (18,883 samples), and (iii) the dataset collected in July 2023 (21,782 samples).**

| Dataset | F1-Score | ROC-AUC | Precision | Recall |
|---|---|---|---|---|
| Hold-out | 0.931 | 0.971 | 0.911 | 0.95 |
| 06/2023 samples | 0.955 | 0.976 | 0.925 | 0.986 |
| 07/2023 samples | 0.953 | 0.943 | 0.919 | 0.99 |

*7.2.2 Fingerprinting.* In our next step, we want to examine how the selection of features based on different approaches affects the results. Any sample within our testing data, as discussed in section §6.2, which generates a fingerprint observed at least once in *Blocklists-Feodo* traffic, is classified as malicious. The calculation of the precision, recall and F1 score is calculated using the *classification_report* function from the python library `metrics`.

Surprisingly, the results reveal a significant deviation. The categories *Exhaustive*, *Predefined*, *ML-Selected* and *All-Possible* achieve precision/recall scores of 0.38/0.94, 0.51/0.69, 0.71/0.59, and 0.64/0.47 respectively. This outcome suggests that while the *All-Possible* category has fewer overlaps compared to *ML-Selected*, the overall overlap count was actually higher leading to more miss-classifications.

The scores highlight the essential role of feature selection in determining the quality of predictions. ATSF [58] uses an emprical strategy of randomly generating *Client Hellos* in order to find the best feature set based on the specific kind of classification each time. Their binary classifier which decides whether a server is a C2 server from a blocklist achieves 99% precision score, with a 35% recall due to the nature of these techniques which rely on exact matching.

### 7.3 Malicious separation

*7.3.1 Machine Learning.* Similarly to binary classification, the multi-class pipeline also succeeds in significantly reducing the feature space of the extracted dataset, from 20K down to a 108 best features, determined by the Lasso regression coefficients. Moreover, the model fine-tuning process accurately identifies the optimal configuration for each of the predefined classification models, as documented in Table 5. In contrast to the binary classification, the pipeline selects the Random Forest model as the most suitable choice, primarily due to its higher average validation F1 score.

Upon selecting the appropriate model, its performance is further evaluated using the hold-out dataset. The chosen configuration for the Random Forest model yields impressive results, achieving a ROC-AUC score of 0.999, F1 score of 0.990, Precision of 0.990 and Recall of 0.990.

*7.3.2 Fingerprinting.* Simultaneously, we use the fingerprinting technique similarly to the binary classification to explore how feature categories behave in a multi-class problem. After updating the new features to the ML-Selected category and considering only labeled traffic, we extract following scores. The categories Exhaustive, Predefined, ML-Selected and All-Possible achieve precision/recall scores of 0.70/0.79, 0.68/0.71, 0.68/0.70, and 0.62/0.28 respectively.

Remarkably, this time, the highest scores are achieved by the *Exhaustive* category, which exclusively leverages the cipher suites selected from the botnet families. Given that each server within this experiment is associated with a distinct botnet family, it is supposed to consistently respond in the same manner. On the other hand, the *All-Possible* category demonstrated lower performance due to the potential issue of overfitting caused by the inclusion of numerous features.

Compared with the binary classification based on fingerprinting in §7.2.2, we see that these techniques work better when servers consistently follow a regular pattern, which makes them particularly suitable for categorizing familiar behaviors. Moreover, since they rely on exact matches with known patterns before, they are optimal for configuration replication discovery. Finally, it is necessary to mention that we do not directly compare them to our machine learning experiments, since we split the dataset differently in order to reflect how each technique would be used in real-life situations.

## 8 ETHICAL CONSIDERATIONS

We contact IP addresses that are advertised as malicious in public blocklists and we do not perform any port scanning. In addition, the communication initiated by our machines did not provoke any reaction from system administrators (e.g., email warnings), as other works mention [46]. Thus, we presume that the presence of our activity in the network is acceptable.

## 9 LIMITATIONS

In this work, we utilize two distinct IP addresses, which are exclusively used for the data collection process. We have not performed any analysis to detect any potential blacklisting from the servers side. For example, it would be possible for servers to detect our activity and respond using fixed TLS Server Hello messages. In the future, we plan to address this limitation.

## 10 CONCLUSION

In this paper, we utilize active TLS fingerprinting techniques in conjunction with a machine learning pipeline to examine whether a server is part of a botnet network and to identify in which specific botnet the server participates. We evaluate 3 machine learning models and 4 distinct feature categories selected with different approaches based on fingerprinting. Our results demonstrate that both machine learning and fingerprinting techniques mutually enhance one another, resulting to higher precision. Finally, upon acceptance of the paper, we plan to make our dataset and selected models configurations publicly available.

As future work, we aim to enrich our TLS fingerprints database with more and different botnets, explore approaches that could help us recognize the randomization of cipher suite vectors and try to recognize servers in the wild. We will perform a more in depth analysis of those server TLS responses specifically to uncommon "TLS Client Hello" configurations. Ultimately, we plan to optimize our fingerprinting techniques through machine learning by refining the utilization of the initial 10 handshakes.

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

# A APPENDIX

## A.1 TLS Handshake

The TLS handshake follows an exchange process between the client and server to verify identities, establish cryptographic keys and ensure communication integrity. Throughout this procedure, a shared secret key is generated and exchanged, along with other relevant parameters essential for establishing a secure channel. Typically, the 3-way handshake is sufficient for collecting and agreeing upon these TLS parameters. However, in some cases, additional requests may be necessary depending on the specific negotiation and extensions used during the process as shown in Figure 4. For example, if the server requests the client's certificate for authentication, an additional exchange ensues where the server prompts for the client's certificate, and the client responds by presenting its certificate. Furthermore, in TLS 1.3, the use of key share extensions requires additional messages for cryptographic negotiation and the Diffie-Hellman key exchange, enhancing the security of the session. Another instance could involve the inclusion of Application-Layer Protocol Negotiation (ALPN) extension, where both client and server exchange messages to harmoniously decide on the application protocol to employ over the secure connection.

## A.2 Cipher Suite Distribution

Figure 5 reveals how different cipher suite scores are spread across the sources. The x-axis of each plot corresponds to the cipher suite scores, measuring the degree of preference or prominence. As the cipher suites score on the x-axis advances, the increasing significance in the negotiation process is apparent. On the y-axis, the cumulative probability is depicted, showcasing the proportion of sources that have cipher scores equal to or less than a given value. For example, a steeper slope on the CDF curve indicates that there is a rapid increase in the proportion of cipher suites with lower scores (as x-axis increases). This phenomenon suggests a prevailing

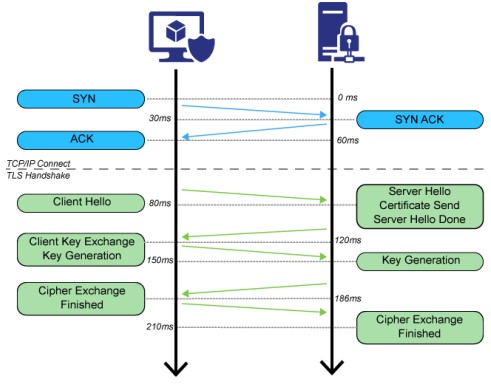

**Figure 4: The TLS handshake [5].**

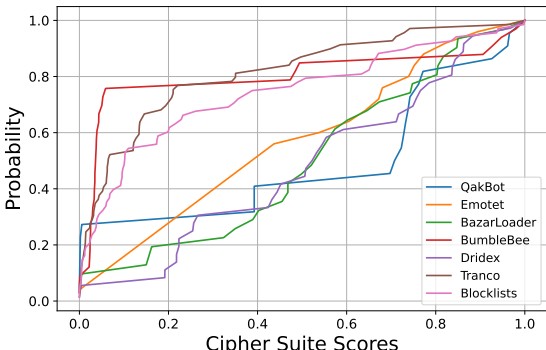

**Figure 5: The cumulative distribution function (CDF) of the normalized cipher suite scores per source**

**Table 7: Confusion matrix of the selected binary model over the hold-out dataset.**

| TN | FP | FN | TP |
|---|---|---|---|
| 108,854 | 517 | 968 | 10,035 |

consensus among sources, where specific cipher suites are consistently prioritized across varying handshakes. This uniformity underscores the stability in their preferences, potentially indicative of standardized security or housekeeping practices. In contrast, a less pronounced slope on the curve indicates a broader distribution of scores. This scenario hints at a diversified selection of cipher suites, pointing to sources that exhibit varying preferences in different contexts. Such behavior could signify adaptability in security protocols or the utilization of diverse encryption strategies, reflecting the nuanced nature of network traffic.

## A.3 Binary classification

*A.3.1 Machine Learning.* Table 7 visually presents a detailed breakdown of the number of accurately predicted samples through a confusion matrix of the Binary classification.

Figure 6 shows the insights about feature values and the effect of those feature values over the final model decision of the Binary classification. More specifically, in the y-axis we can see the most significant features which have the biggest influence over the model decision. Each of the presented features have a range of values, which are represented between blue (low feature values) and red (high feature values) colors. Furthermore, the x-axis represents the impact of the particular feature value over the final model decision (SHAP value). The higher SHAP values push the model decision towards the positive decision, in our case represented as the malicious sample prediction, and lower values push the model towards the negative decision (benign class). For example, samples with low value of port number will push the model towards predicting a server as benign, and the higher port number samples will be classify a server as malicious. Additionally, the dot cluster thickness provides information about the distribution of tested values. In the case of the port number feature, it is clear that most

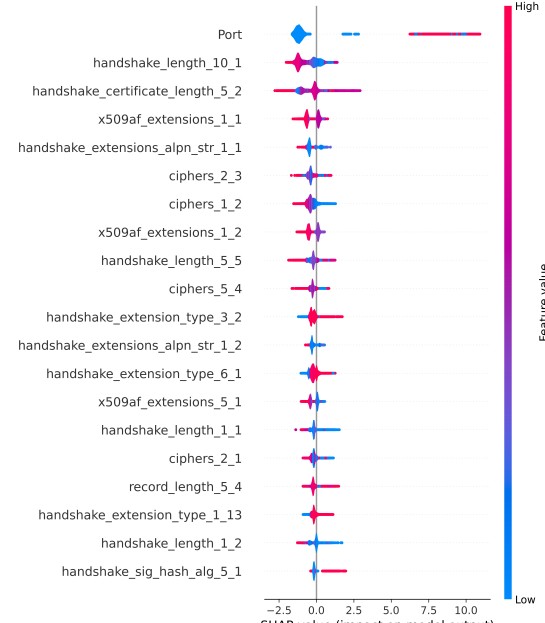

Figure 6: SHAP decision explanation of the binary classification model. Illustrates the variances among the top 20 significant features and their influence on the final model's decision (i.e. handshake_length_10_1 corresponds to the handshake length extracted from the 10th handshake and the 1st *Server Hello*).

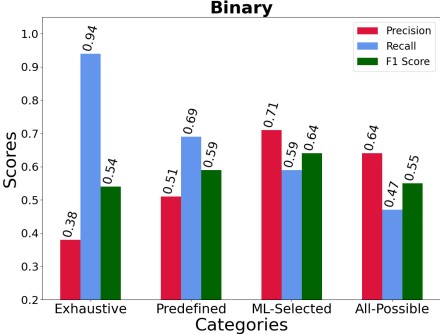

Figure 7: The performance of each category in Binary classification based on fingerprinting

of the samples have low values since our dataset is imbalanced and a higher percentage of tested samples belong to the benign class. Furthermore, as it is shown in Figure 6 the separation and the effect of feature values is not always binary since in some cases machine configuration may utilize low port number but additional features will lead to a classification decision towards malicious.

*A.3.2 Fingerprinting.* Figure 7 shows the precision, recall and F1 score that achieved each category during the Binary classificaiton process based on TLS Fingerprinting.

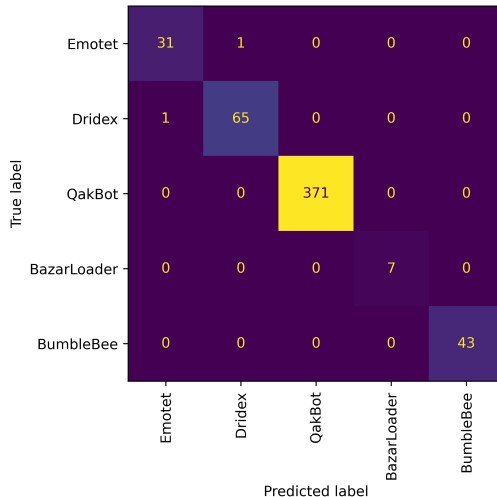

Figure 8: Confusion matrix of our multi-class classification model over the hold-out dataset portion.

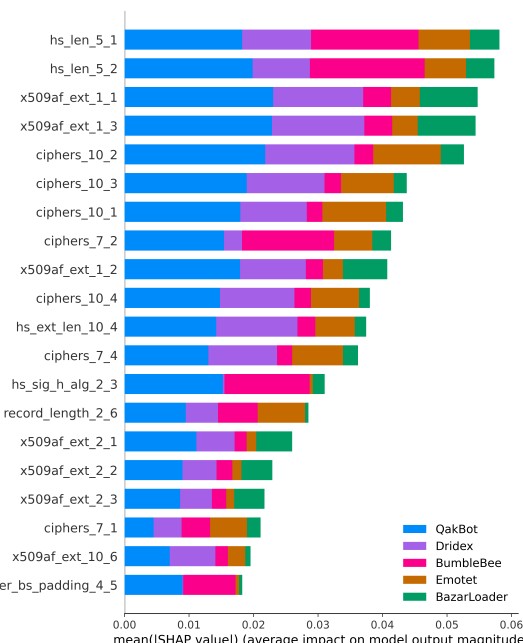

Figure 9: SHAP explanation of the secondary model based on multi-class classification. This figure illustrates the average impact of the top 20 features over the final model's decision.

## A.4 Multi-Class classification

*A.4.1 Machine Learning.* Figure 8 shows the detailed predictions for each class during the Multi-Class classification.

Similar to the binary classification, in the case of multi-class classification, we also utilize the SHAP explainability method to extract additional insights from our developed method concerning prediction decisions (Figure 9). Unlike the previous explainability method

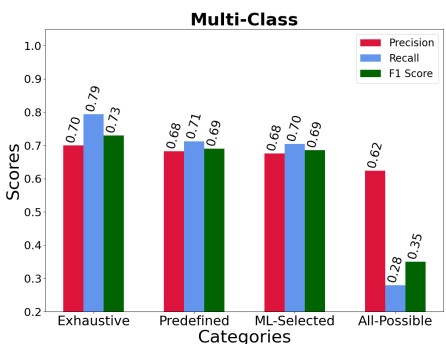

**Figure 10: The performance of each category in Multi-Class classification based on fingerprinting**

(binary classification), the current figure illustrates the mean impact of the most important features overall for class differentiation. In this representation, features are displayed on the y-axis, and the impact of each feature on a specific class can be discerned through the colored bars and their sizes. For instance, the feature *hs_len_5_1* has a significant impact on the QakBot and BumbleBee categories, while its impact is much lower on the Dridex and BazarLoader classes. Furthermore, certain features appear to have no impact on

certain categories at all, such as the feature *ber_bs_padding_4_5* for the Dridex category. The explainability presented offers general insights into the patterns captured by the developed classification model for each botnet category.

Moreover, Figure 11 provides the SHAP multi-class model explainability for each specific botnet category. The presented figures illustrate the top 20 features for each botnet along with their decision distribution.

*A.4.2 Fingerprinting.* Figure 10 shows the precision, recall and F1 score that achieved each category during the Multi-Class classification process based on TLS Fingerprinting.

## A.5 Parameters Extraction

Lastly, Table 8 contains the parameters extracted from each packet capture file from our database. Each parameter derived from each individual *Client Hello* and it is presented with the corresponding name from our source code alongside with a small description. We select a wide range of options available for feature selection. This, combined with the exhaustive approach allow us to exponentially increase the total number of parameters. We aim to generate an extensive collection of 20K potential features and let machine learning choose the optimal ones. Since manually testing every possible combination is impractical, even impossible, we leveraged a machine learning pipeline for accomplishing this.

**Table 8: The TLS parameters extracted from each *Server Hello* with a brief description**

| Parameter | Description |
| --- | --- |
| versions | TLS protocol version |
| ciphers | Cipher suite |
| record_length | Length of the TLS record |
| handshake_length | Total length of the handshake message |
| handshake_extensions_length | Length of the handshake extensions field in bytes |
| handshake_extension_type | Type of the handshake extension included in the negotiation |
| handshake_extensions_reneg_info_len | Length of the renegotiation information field |
| handshake_extensions_ec_point_formats_length | Length of the elliptic curve point formats list in the handshake extensions |
| handshake_extensions_alpn_len | Length of the Application-Layer Protocol Negotiation |
| handshake_extensions_alpn_str_len | Length of the ALPN protocol string in the handshake extensions |
| handshake_extensions_alpn_str | ALPN protocol string indicating the selected application protocol |
| handshake_fragment_count | Number of fragments used for the handshake |
| handshake_certificate_length | Total length of the X.509 certificate chain |
| x509af_signedcertificate_element | Element containing the signed certificate in the X.509 certificate structure |
| x509af_version | Version number of the X.509 certificate |
| x509af_serialnumber | Serial number of the X.509 certificate |
| x509af_signature_element | Element containing the signature in the X.509 certificate |
| x509af_algorithm_id | Algorithm identifier used for the signature in the X.509 certificate |
| x509af_issuer | Issuer of the X.509 certificate |
| x509if_rdnsequence | RDN sequence in the X.509 certificate |
| x509if_rdnsequence_item | Individual item in the RDN sequence of the X.509 certificate |
| x509if_id | Identifier for the X.509 certificate |
| x509sat_countryname | Country name field in the X.509 Subject Attribute Type |
| x509sat_directorystring | Directory string in the X.509 Subject Attribute Type |
| x509sat_utf8string | UTF-8 string in the X.509 Subject Attribute Type |
| x509af_validity_element | Element containing the validity period of the X.509 certificate |
| x509af_notbefore | Certificate validity start date in the X.509 certificate |
| x509af_notafter | Certificate validity end date in the X.509 certificate |
| x509af_subject | Subject of the X.509 certificate |
| x509af_rdnsequence | RDN sequence in the subject of the X.509 certificate |
| x509af_subjectpublickeyinfo_element | Element containing the subject public key info in the X.509 certificate |
| x509af_algorithm_element | Element containing the algorithm info in the X.509 certificate |
| x509af_extensions | Extensions included in the X.509 certificate |
| x509af_extension_id | Identifier for the extensions in the X.509 certificate |
| x509ce_authoritykeyidentifier_element | Element containing the authority key identifier in the X.509 certificate |
| x509ce_basicconstraintssyntax_element | Element containing basic constraints info in the X.509 certificate |
| x509ce_ca | Basic constraints indicating if the certificate is a CA or an end-entity |
| x509af_algorithmidentifier_element | Element containing the algorithm identifier in the X.509 certificate |
| ber_bitstring_padding | Padding used for the BER-encoded bitstrings |
| handshake_server_curve_type | Elliptic curve type used by the server |
| handshake_server_named_curve | Named curve used by the server |
| handshake_server_point_len | Length of the server's elliptic curve point |
| handshake_sig_hash_alg | Signature hash algorithm used |
| handshake_sig_hash_hash | Hash value used in the signature |
| handshake_sig_hash_sig | Signature value used |
| handshake_sig_len | Length of the signature used |

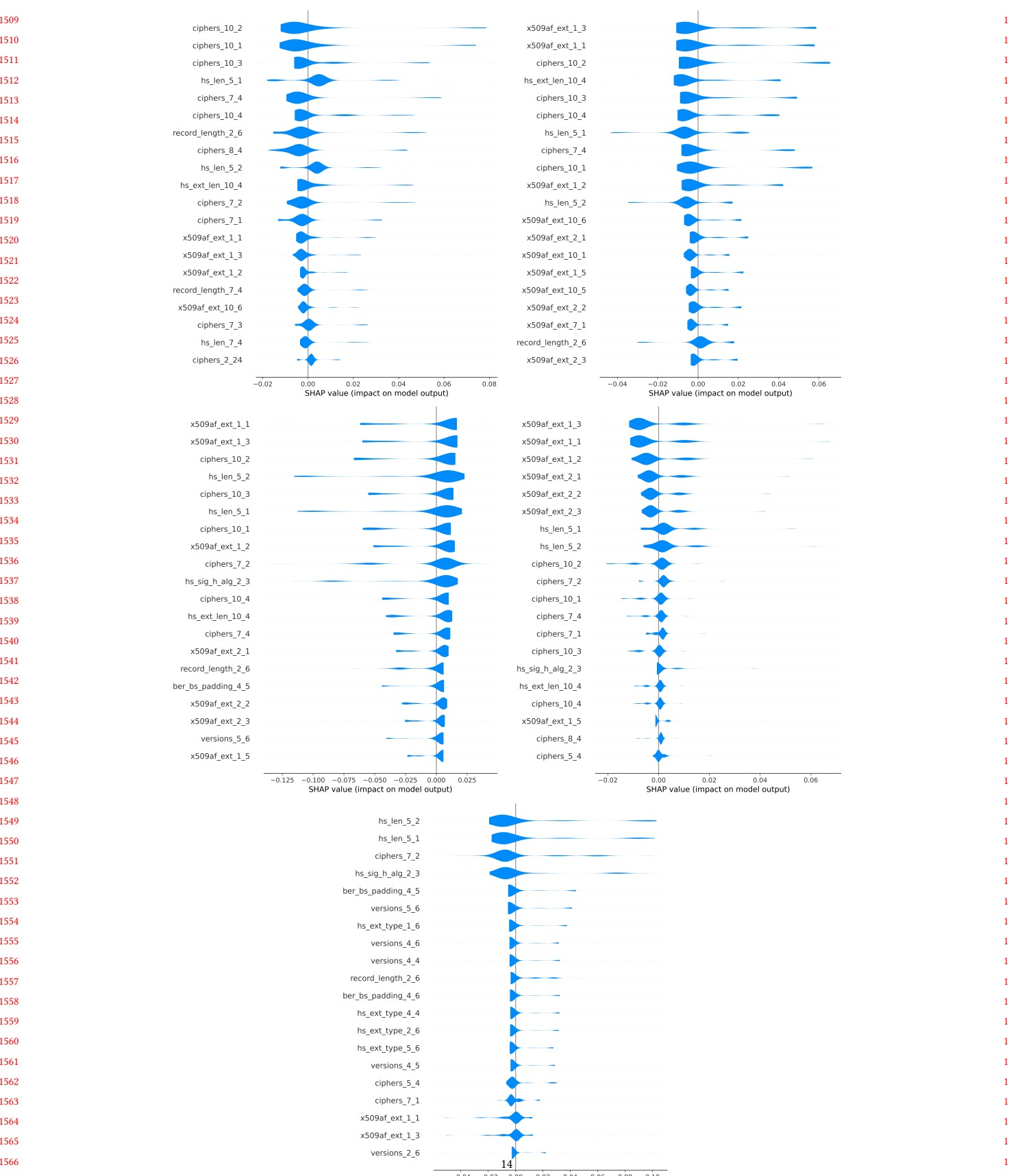

**Figure 11: SHAP decision explanation (abbreviations are used) of the multi-class classification model (i.e. ciphers_10_2 corresponds to the cipher suite extracted from the 10th handshake and the 2nd *Server Hello*). Each figure presented in an order from top left to the bottom right corresponds to the following botnets: QakBot, Dridex, BumbleBee, Emotet and BazarLoader**