# OpenReview forum: "Fingerprinting the Shadows: Unmasking Malicious Servers with Machine Learning-Powered TLS Analysis"
_ACM.org/TheWebConf/2024/Conference — TheWebConf24 Oral_

### Official Review · Reviewer_RP9R · 2023-11-17

**Novelty:** 4
**Technical Quality:** 5

**Review:**

This work aims to classify malicious servers and their particular categories using machine learning. The machine learning features were learned from server_hello messages, as responses to fabricated client_hello messages, while the later ones are used for forcing servers to select possibly different cipher suites for TLS connection. The experiments were performed on websites of Tranco 10K list, Feodo, and Blocklists. Overall, the results look promising but maybe incomplete.

Pros:
1. Using cipher suites and other parameters, e.g., those about the certificate, for the classification of malicious servers, which has not yet been studied.
2. A proper selection of benign and malicious datasets.

Cons:
1. [57] is highly likely the baseline to compare with, which is missing in this work, since [57] adopted the same strategy to obtain parameters from TLS handshake (although for a different purpose).
2. When saying "no prior work has delved into the use of machine learning models for ...Our research seeks to address this limitation..." in line 547, it is unclear about why not using machine learning becomes a limitation.
3. In line 317 it is mentioned to "evaluate the effectiveness of various state-of-the-art approaches" but I could not find such an evaluation in the paper.
4. It is unclear about how the unlabeled Blocklists dataset served experiments regarding machine learning and fingerprinting in Sections 6 and 7, as the paper stated "Blocklists ... refers to these unlabled lists" in line 279, while no manual labeling was mentioned in the paper.
5. It is better to provide more details about the "additional layer of complexity" when using machine learning in line 181.
6. Why does it use 10 but not 11 or 9 client_hello messages in line 289?
7. In Section 4.3, how large the sets of incomplete, disrupted, and those don't match up or follow the rules are? Can they serve fingerprinting? In browser fingerprinting, no fingerprint can also be fingerprinted.
8. "20,000 features" in line 400 should be "20,700 features".
9. Why does it not just use the hash, but perform a concatenation on the hash, in lines 670 to 673?

**Questions:**

The cons above.

**Reviewer Confidence:**

4: The reviewer is certain that the evaluation is correct and very familiar with the relevant literature

**Scope:**

3: The work is somewhat relevant to the Web and to the track, and is of narrow interest to a sub-community

---

### Official Review · Reviewer_3dyX · 2023-11-20

**Novelty:** 5
**Technical Quality:** 5

**Review:**

The paper presents a study on the fingerprintability of server configurations with respect to the TLS protocol, to classify them as either benign or malicious. The main contribution is the exploration of different techniques, including existing ML-based ones, on a large dataset. For the ML part, the authors apply an existing pipeline with different sets of features that they extracted through active probing, instead of passive.

Pros:
* Large dataset collected in a reproducible way
* Comparison between ML-based and fingerprinting- (i.e., signature-) based techniques
* Sound methodology and clear description
* Ground truth labelling clearly explained and justified

Cons:
* Feature selection for fingerprinting method is not sufficiently described (Sec. 6.2): "Based on current approaches we create 4 distinct combinations of features.".
* Missing details about experimental machine measurements (how long did the training take? How much memory was needed? etc.)
* Lack of comparison of results with related work (for the fingerprinting-based techniques). As this work basically compares existing techniques (aside from the new ML features), it should contain the comparison with the prior works and explanation of their potentially differing results.
* Unnecessary repetitions in Sec. 4.1 from this section itself and from the previous one.

**Questions:**

None.

**Reviewer Confidence:**

3: The reviewer is confident but not certain that the evaluation is correct

**Scope:**

4: The work is relevant to the Web and to the track, and is of broad interest to the community

---

### Official Review · Reviewer_qykq · 2023-11-22

**Novelty:** 4
**Technical Quality:** 5

**Review:**

This paper proposes a new method for extracting server classification and fingerprinting features. Besides, this paper leverages a range of active TLS fingerprinting techniques to examine server behavior. The authors implement binary classification systems to label benign/malicious servers and multi-class classification models to identify specific botnet families. The evaluation part shows the effectiveness of the proposed methods.

Pros

             1. Well-structured

             2. Important topic

             3. Detailed evaluation of the proposed methods

Cons

             1. Limited novelty

             2. Lack of comparison with existing works

             3. Some parts are not clear

**Questions:**

1. The novelty of this paper is not clear. The novelty of this paper is mainly located in adopting the mean number of successful handshakes and the selected cipher suites as the feature for classification and fingerprinting, which is not enough. The authors are supposed to demonstrate their novelty more clearly and pertinently.

2. This paper lacks comparisons with existing works, such as ATSF, JARM, and DissecTLS. Quantitative comparisons with these works regarding accuracy, efficiency, recall, etc., are needed for better evaluation.

3. Some parts of this paper need to be clarified. First, the reasons for selecting the parameters in Table 8 are not clear. Were these choices based on some previous works? Besides, the details of the selected features in Table 3 are not presented.

4. I am curious about why the choice of cipher suites can be used to identify benign and malicious servers. I would appreciate it if the authors provided some intuitive explanations.

5. Another problem is why the features formalized by a hash function could be used for classification, as the hashed data will vary significantly with slight differences in the inputs.

**Ethics Review Description:**

No issues found.

**Reviewer Confidence:**

3: The reviewer is confident but not certain that the evaluation is correct

**Scope:**

4: The work is relevant to the Web and to the track, and is of broad interest to the community

---

### Official Review · Reviewer_7oou · 2023-11-24

**Novelty:** 5
**Technical Quality:** 5

**Review:**

This research aims to identify servers that are used for malicious purposes such as C&C by leveraging TLS fingerprinting through machine learning techniques. Notably, they utilize active probing of the servers using crafted TLS Client Hello requests to monitor the gather the difference in the responses sent by the server. Later, features identified from these responses per server are used to train the ML models. They obtain the ground truth for benign servers by probing servers from Tranco’s top 10K ranked domains. Through this methodology, their approaches prove successful in classifying benign and malicious servers. Further, they also report that such TLS-based features can be successfully used to identify servers of different botnet families.

Strengths:

1) Interesting insights on the difference in configuration between benign and malicious servers
2) Uses a large dataset of benign and malicious samples to test their approach. Dataset on making it public would prove useful to the community.
3) Experiments, feature selection and model selection are conducted methodically.

Weaknesses:

1) Focuses only on C&C servers
2) Detection approach is not robust enough and can be evaded by modifying server configurations to mimic legitimate servers

**Questions:**

1) Provide more clarifications on the data collected from blocklists. Specifically, are these IPs solely associated with Botnet servers
2) While calculating overlaps of IPs between databases, it would be useful to identify how many of these malicious IPs are owned by popular services and draw relations (if any) between the overlaps between these malicious servers and the benign servers
3) Provide more discussion on how robust is this classification if attackers choose to mimic legitimate behavior.


**This is to acknowledge that I have read the rebuttal.**

**Reviewer Confidence:**

4: The reviewer is certain that the evaluation is correct and very familiar with the relevant literature

**Scope:**

4: The work is relevant to the Web and to the track, and is of broad interest to the community

---

### Official Review · Reviewer_dteh · 2023-11-25

**Novelty:** 4
**Technical Quality:** 2

**Review:**

## Summary
This paper introduces a machine learning (ML) based method to identify malicious servers through TLS handshakes. The authors utilize DissectTLS to examine a set of popular websites and blocklisted sites, aiming to detect unique features and fingerprints for training a classifier. They further develop a multi-class classifier to distinguish malware families based on server characteristics. The evaluation encompasses 10,000 well-known sites and a number of blocklisted sites, the exact count of which is not specified.
## Strengths
  - The use of datasets from various periods enhances the robustness of the evaluation.
  - The research addresses a compelling problem, offering a potential method for detecting malicious servers through active scanning.
  - The technical integration of TLS cipher information is noteworthy and has broader applications in areas like vulnerability analysis and compliance.
## Weaknesses
  - The paper could benefit from a clearer presentation, particularly in explaining ML feature extraction and lacking practical examples.
  - The methodology for transformation and parameter selection is vaguely described, lacking an in-depth discussion of the details.
  - Standard feature transformation processes are mentioned but fail to offer insightful details about the experimental setup.
  - While feature analysis is covered, it lacks direct references to figures in the appendix.
  - The method of differentiating between successful probing and disrupted (timed out) cases is unclear.
  - Validation of blocklists is absent; these often contain false positives due to CDNs and shared hosting, raising questions about handling such scenarios.
  - The results, particularly in section 5.1 regarding response analysis, are speculative and lack concrete empirical backing.
  - The rationale for excluding the analysis of the last handshake from the scope is not explained.
  - In Figure 2 regarding blocklists, the absence of IP address validation casts doubt on the reliability of the results.
  - Concerning fingerprinting techniques:
    - The rationale behind not employing PCA or a feature ranking approach for selecting crucial fingerprint features is not addressed.
    - The fingerprinting technique itself is not validated, leading to uncertainty about the stability and distinctiveness of these fingerprints.
    - The reproducibility of fingerprints upon repeated TLS extraction is questionable.
    - The problem of overlap in fingerprints between blocklisted IPs and Tranco (clean) sites, as seen in Figure 3 and Section 7.2.2, is a significant concern.

Overall, the paper has promising early results, but I do not think it is ready for publishing. Additional evaluation and validation is required. Specifically, I recommend the authors to focus on:
 1. Characterizing the IP addresses from Tranco and the blocklist (ASNs, Orgs, Geo distribution, etc.) gives us a sense of the data representation. Additionally, including top sites from various regions across the world would improve the confidence in the results.
 2. Validate the uniqueness and reproducibility of the fingerprints.
 3. Provide explicit numbers regarding the dataset, how many IP addresses, what was filtered, what the overlap between datasets, and how the final training and testing dataset stems from the initial IPs.
 4. Investigation into the last handshake anomaly and providing some explanation of why these factors happen, how they can impact the collected dataset, and how to mitigate it.
 5. Focus on feature ranking and what are the key factors that differentiate malicious from benign. What key features differentiate malware families?

**Questions:**

Did the authors validate their source of top sites and blocklists?
Did the authors validate the reproducibility of the TLS feature extraction? Is it deterministic?
Did the authors validate their fingerprinting hash? How does it impact the results?
What factor limited the use of only Fedeo list? Why not also include Threatfox and broader list from abuse.ch?

**Ethics Review Description:**

the authors discuss ethical considerations, but not required for this study.

**Reviewer Confidence:**

3: The reviewer is confident but not certain that the evaluation is correct

**Scope:**

4: The work is relevant to the Web and to the track, and is of broad interest to the community

---

### Decision · Program_Chairs · 2024-01-22

**Decision:**

Accept (Oral)

**Comment:**

### Meta Review:

 **Pros:**
 1. **Novelty in Server Classification:** The paper introduces a novel approach to classify servers as benign or malicious using machine learning features extracted from TLS handshake messages, focusing on cipher suites and other parameters.
 2. **Proper Dataset Selection:** The authors utilize a proper selection of benign and malicious datasets, including websites from Tranco 10K list, Feodo, and Blocklists, enhancing the credibility of their experiments.
 3. **Reproducibility:** The paper emphasizes reproducibility, and the large dataset collection is acknowledged as a strength.
 4. **Methodological Clarity:** The methodology for active TLS fingerprinting is well-described, and the paper maintains a clear structure.

 **Cons:**
 1. **Missing Baseline Comparison:** The absence of a baseline for comparison, particularly with [57], which employs a similar strategy for obtaining parameters from TLS handshake, is noted as a significant drawback.
 2. **Unclear Limitations Statement:** The paper mentions that "no prior work has delved into the use of machine learning models," but it is unclear why not using machine learning is considered a limitation.
 3. **Lack of Expected Evaluation:** The promise to "evaluate the effectiveness of various state-of-the-art approaches" is noted in the review, but the evaluation is not found in the paper, raising concerns about clarity.
 4. **Dataset Uncertainty:** The handling of the unlabeled Blocklists dataset raises concerns, especially when the paper refers to it as "unlabeled lists" but does not describe the manual labeling process.
 5. **Lack of Detail:** Some aspects require additional details, such as the "additional layer of complexity" in using machine learning, the choice of using 10 client_hello messages, and the size of sets in Section 4.3.
 6. **Inconsistencies in Figures:** Inconsistencies are noted in figures, such as the mention of "20,000 features" when it should be "20,700 features" and the choice of concatenation on the hash in lines 670 to 673.

 **Suggestions:**
 1. **Baseline Comparison:** Include a baseline comparison, especially with [57], to provide a more comprehensive evaluation of the proposed approach.
 2. **Clarify Limitations:** Provide a clearer explanation of why not using machine learning is considered a limitation.
 3. **Ensure Expected Evaluation:** Address the statement about evaluating state-of-the-art approaches by incorporating the expected evaluation in the paper.
 4. **Clarify Dataset Handling:** Clearly explain the handling of the unlabeled Blocklists dataset and how it serves experiments regarding machine learning and fingerprinting.
 5. **Provide Additional Details:** Offer more details about the "additional layer of complexity" when using machine learning, the choice of using 10 client_hello messages, and the size of sets in Section 4.3.
 6. **Check Figure Consistency:** Address inconsistencies in figures, such as the mention of "20,000 features," and provide accurate information.

 ### Conclusion:
 The paper introduces a novel approach to classify servers as benign or malicious through machine learning features extracted from TLS handshake messages. While the methodology and dataset selection are acknowledged strengths, the lack of a baseline comparison, unclear limitations statement, missing expected evaluation, dataset handling uncertainties, and inconsistencies in figures are significant drawbacks. Addressing these issues, providing additional details, and ensuring figure consistency are crucial for enhancing the paper's overall quality.

 ---